# RISK-SENSITIVE AGENT COMPOSITIONS

**Guruprerana Shabadi, Rajeev Alur**
University of Pennsylvania
{shabadi,alur}@seas.upenn.edu

## ABSTRACT

From software development to robot control, modern agentic systems decompose complex objectives into a sequence of subtasks and choose a set of specialized AI agents to complete them. We formalize agentic workflows as directed acyclic graphs, called agent graphs, where edges represent AI agents and paths correspond to feasible compositions of agents. Real-world deployment requires selecting agent compositions that not only maximize task success but also minimize violations of safety, fairness, and privacy requirements which demands a careful analysis of the low-probability (tail) behaviors of compositions of agents. In this work, we consider risk minimization over the set of feasible agent compositions and seek to minimize the value-at-risk and the conditional value-at-risk of the loss distribution of the agent composition where the loss quantifies violations of these requirements. We introduce an efficient algorithm which traverses the agent graph and finds a near-optimal composition of agents. It uses a dynamic programming approach to approximate the value-at-risk of agent compositions by exploiting a union bound. Furthermore, we prove that the approximation is near-optimal asymptotically for a broad class of practical loss functions. We also show how our algorithm can be used to approximate the conditional value-at-risk as a byproduct. To evaluate our framework, we consider a suite of video game-like control benchmarks that require composing several agents trained with reinforcement learning and demonstrate our algorithm's effectiveness in approximating the value-at-risk and identifying the optimal agent composition.

## 1 INTRODUCTION

Modern agentic workflows orchestrate specialized AI agents to tackle complex tasks demanding a diverse range of skills. Typically, a high-level planning agent decomposes the main task into a sequence of subtasks and selects appropriate worker agents to execute them. These worker agents can be powerful generalist models, such as large language models (LLMs) and vision-language models (VLMs), or control policies trained with reinforcement learning or imitation learning. Such systems have demonstrated considerable success in automating tasks across various domains, including software development (Zhang et al., 2025a; Niu et al., 2025; Hu et al., 2025), complex information retrieval (Zhang et al., 2025b), scientific discovery (Gridach et al., 2025), and robot control (Ichter et al., 2022; Feng et al., 2025; Yang et al., 2024; Zhao et al., 2025).

In addition to maximizing task success, deploying agentic systems in the real world requires minimizing various forms of risk. The stochastic nature of both environment and agents means that we have to rigorously analyze low-probability but high-consequence tail behaviors that emerge when multiple agents interact. Our work follows a rich body of literature in risk-sensitive planning and control that dates back to the 1970s (Howard and Matheson, 1972; Whittle, 1981) where the objective is to optimize a risk measure of the loss (or reward) distribution as opposed to the expectation. A risk measure is a function that maps a loss distribution to a real number and takes into account attributes like the variance and the tail of the distribution. Some examples include the value-at-risk (VaR) which corresponds to the tail quantile and conditional value-at-risk (CVaR) which is the expected tail loss. In this work, we consider loss functions that quantify violations of requirements such as safety, fairness, and privacy.

In our formalization, we represent an agentic workflow by an *agent graph* (example in Figure 1) which is a directed acyclic graph in which edges correspond to agents (denoted $\pi_1, \ldots, \pi_4$) and

paths correspond to feasible agent compositions that achieve the overall objective. In Figure 1, we have two compositions of agents to pick from: $p_A := \pi_1 \to \pi_3$ and $p_B := \pi_2 \to \pi_4$. Finally, the risk associated with an agent's behavior is quantified with a real-valued loss function $L_i : T_i \to \mathbb{R}$ that takes as input a trace of execution of the corresponding agent $\pi_i$.

**Example 1: DroneNav.** Consider four control agents $\pi_1, \ldots, \pi_4$ that have been trained to navigate a drone between the rooms $S \xrightarrow{\pi_1} A$, $S \xrightarrow{\pi_2} B$, $A \xrightarrow{\pi_3} F$, and $B \xrightarrow{\pi_4} F$. We define loss functions $L_i : T_i \to \mathbb{R}$ for each agent that capture safety requirements. They take as input the trajectory from a rollout of a policy and return the negative minimum distance between the obstacles in the rooms and the trajectory. Now, consider the task of navigating a drone from room $S$ to the target room $F$ which can be achieved by two paths: one passing through room $A$ and another through room $B$. This can be represented by the agent graph in Figure 1 where we have the two

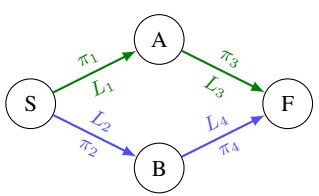

Figure 1: An agent graph

compositions of agents to choose from. The loss functions of the compositions of agents $p_A$ and $p_B$ correspond to the negative minimum distance between the obstacles and the *sequence* of trajectories of the agents in each composition. In particular, for the composition $p_A$, the loss function is denoted $L_{p_A} : T_1 \times T_3 \to \mathbb{R}$ and is given by $L_{p_A}(t_1, t_3) := \max(L_1(t_1), L_3(t_3))$ where $t_1$ and $t_3$ are trajectories from rollouts of $\pi_1$ and $\pi_3$ respectively. $L_{p_B}$ is symmetrically defined.

**Example 2: Information retrieval.** Suppose we have access to an LLM agent that we are using in an information retrieval and processing task. In this setting, a natural requirement is to minimize the amount of hallucinated information, i.e., data that is not taken from data sources or the LLM context. We define the loss function to be a metric that quantifies the amount of hallucinated information in the outputs which can be a score produced by an LLM-as-a-Judge. Next, consider the task of constructing an agentic system that takes as input the name of a country and produces a graph of the evolution of literacy rates in the country over a period of five decades. Also, suppose we have two data sources—UNESCO and the World Bank—from which the system needs to fetch relevant pages and then write a Python script to generate the plot. This can be represented by the agent graph in Figure 1 where $\pi_1$ and $\pi_2$ represent LLM agents tasked with retrieving relevant text data from the two sources, and $\pi_3$ and $\pi_4$ represent agents that write the script to produce the plot from the data. Yet again, the loss of the composition of agents is given by $L_{p_A}(t_1, t_3) := \max(L_1(t_1), L_3(t_3))$ which now corresponds to the maximum amount of information hallucinated by either of the two agents $\pi_1$ or $\pi_3$ in their corresponding outputs $t_1$ and $t_3$.

As noted in these examples, when the loss of each agent quantifies violations of safety, fairness, and privacy requirements, the loss of a composition of agents—the quantity we want to minimize—is the maximum of the losses incurred by each agent. In contrast, existing literature in risk-sensitive planning usually considers the *cumulative* loss (Ahmadi et al., 2021; Kretínský and Meggendorfer, 2018) which is the natural choice when the losses represent costs.

Given an agent graph, our objective is to find the path representing an agent composition that minimizes the *value-at-risk*, denoted $\mathrm{VaR}_\alpha$, which is the $(1 - \alpha)$-quantile of the loss distribution of the agent composition for a user-defined risk budget $\alpha > 0$. In other words, minimizing $\mathrm{VaR}_\alpha$ minimizes a constant $\ell^* > 0$ such that with probability at least $1 - \alpha$, each agent in the chosen composition of agents incurs a loss of at most $\ell^*$. We also consider another risk measure known as the *conditional value-at-risk* (also known as expected shortfall), denoted $\mathrm{CVaR}_\alpha$, which is defined as the expected loss in the worst $\alpha$ fraction of the loss distribution. In the rest of this work, we assume only black-box access to the agents and focus on estimating these risk measures through sampling.

A straightforward approach to minimizing the value-at-risk is to consider each feasible agent composition, estimate its tail quantile, and pick the optimal composition. However, this method is inefficient since the graph can have an exponential number of compositions as a function of the number of agents. We overcome this hurdle with an efficient algorithm that scales polynomially in the number of agents and incrementally builds a near-optimal path while traversing the agent graph. At its heart is a quantile estimation method that approximates the value-at-risk of a composition of agents by dividing the risk budget $\alpha$ amongst the agents and applying a union bound. It then finds the optimal allocation of the risk budget over a discretized set of budgets using a dynamic programming algorithm. We prove that this approximation is near-optimal asymptotically when the losses of agents

are given by independent random variables. This independence assumption holds when the losses are functions of the agent behavior and do not directly impact task accuracy or performance. Indeed, safety, fairness, and privacy are often properties of the behavior and are orthogonal to task success. Further, since the conditional value-at-risk can be approximated as the average of tail quantiles, we show how it can be recovered as a byproduct of our algorithm.

We implement our algorithm and test its performance on a series of reinforcement learning (RL) environments. Here, we look at long-horizon tasks that require composing several agents trained with RL to complete them. We consider two tasks—safety in navigation and fair resource consumption—and design suitable loss functions that capture these requirements. We confirm empirically that our algorithm produces tight approximations of the value-at-risk and the conditional value-at-risk thereby finding optimal agent compositions. Further experiments show that our algorithm is robust to a reasonable amount of correlation between agent losses.

In summary, our contributions consist of the following:

1. We formalize the risk minimization objective for agentic workflows as the problem of finding an agent composition that minimizes the value-at-risk of the loss quantifying violations of safety, fairness, and privacy requirements (Section 2).

2. Then, we introduce an efficient algorithm that approximates this optimization problem with a quantile estimation method using the union bound and dynamic programming. Furthermore, we prove that this approximation is near-optimal asymptotically under independence assumptions. (Section 3)

3. Finally, we evaluate our algorithm on compositional reinforcement learning benchmarks using two tasks: safety-critical navigation and fair resource consumption. We design appropriate loss functions for these tasks and demonstrate that our algorithm successfully identifies optimal agent compositions while providing tight approximations of their risk measures (Section 4).

## 1.1 RELATED WORK

Our work builds upon a large body of literature on automated planning and control under uncertainty (Ghallab et al., 2016). Traditionally, the planning problem is modeled as a Markov decision process (MDP) over which the expected (un)discounted cumulative loss is minimized using algorithms like value iteration or reinforcement learning. Towards designing risk-aware agents, recent works have instead considered the optimization of risk measures that look at the tail of the loss distribution instead of the expected loss. Ahmadi et al. (2021); Kretínský and Meggendorfer (2018) study risk minimization for reachability and mean-payoff objectives in MDPs, while Bastani et al. (2022); Wang et al. (2023); Greenberg et al. (2022); Choi et al. (2021) develop risk-sensitive reinforcement learning algorithms. These works consider risk measures that fall into the class of coherent risk measures (Artzner et al., 1999) that include the conditional value-at-risk and the entropic value-at-risk. Risk-sensitive objectives have also been considered in the stochastic control literature. Methods for risk-sensitive linear, quadratic, and Gaussian control (Whittle, 1981) have been developed along with more recent methods for non-linear model predictive control (Nishimura et al., 2020). We also refer to Wang and Chapman (2022) for a comprehensive survey of this topic. However, all of these works stand in contrast with our framework where we want to minimize a risk measure of the *maximum* incurred loss instead of the cumulative loss.

Related lines of research in robotics include hierarchical reinforcement learning (Jothimurugan et al., 2021; Dalal et al., 2024; Zhou et al., 2024; Huang et al., 2023; Lin et al., 2024) and task and motion planning (TAMP) (Kaelbling and Lozano-Pérez, 2011) in which long-horizon control objectives are decomposed into subtasks that are carried out by motion planners or other low-level controllers trained with RL or imitation learning. Similar frameworks have been proposed for automatic generation of agentic workflows to automate software development tasks (Niu et al., 2025; Zhang et al., 2025a; Hu et al., 2025) that sequence LLM-agents to complete smaller subtasks. Agentic systems have also been proposed for the tasks of complex information retrieval (Zhang et al., 2025b) and scientific discovery (Gridach et al., 2025). These methods typically optimize the overall task success probability and do not consider risk minimization.

## 2 RISK MINIMIZATION OVER AGENT GRAPHS

In this work, an *agent* is any machine learning model $f : X \to \mathfrak{D}(T \times Y)$ where $X$ is the input domain, $T$ is the set of computational traces capturing the set of agent behaviors, $Y$ is the output domain, and $\mathfrak{D}(T \times Y)$ is the set of probability distributions over $T \times Y$. For example, for an agent controlling a robot, $X$ is the set of initial states, $T$ is the set of trajectories of the robot in its environment, and $Y$ is the set of final states. In the case of a language model, $X$ is the set of prompts, $Y$ is the set of final responses, and $T$ is the set of intermediate tokens used to produce the final response. This can be the Chain-of-Thought reasoning traces, or interactions with a tool like retrieval from the web or running code. We consider a distribution over traces and outputs, as opposed to a single output, to enable modeling stochastic agent behaviors along with interactions with a stochastic environment. We only assume sampling access to the distribution over outputs and denote $(t, y) \sim f(x)$ for sampling a trace-output pair $(t, y)$ from the agent $f$ for the input $x$.

We quantify the risk associated with an agent's trace by a loss function $L : T \to \mathbb{R}$. This formulation allows us to decouple risk, which depends on the agent's overall behavior, from task accuracy, which only depends on the final output. Thus, the loss function can encode desiderata of the agent's behavior like safety, privacy, and fairness. For example, in the DroneNav environment from Example 1, the loss function quantifies how safe a control policy is on a trajectory of the drone.

Observe that given a distribution over the inputs $\mathcal{D}(X)$, the agent $f$ induces distributions $\mathcal{D}_f(T)$ and $\mathcal{D}_f(Y)$ over the set of traces and outputs respectively. We can sample $t \sim \mathcal{D}_f(T)$ or $y \sim \mathcal{D}_f(Y)$ by first sampling $x \sim \mathcal{D}(X)$, then sampling $(t, y) \sim f(x)$, and dropping one of the two components. Evaluating the loss on the distribution of traces gives us a distribution over the losses the agent incurs. Since we are concerned with risk minimization, we focus on analyzing tail behaviors of the loss distribution. In particular, we are interested in optimizing the following risk measures: the *value-at-risk* at level $\alpha \in (0, 1)$, denoted $\text{VaR}_\alpha$ and the *conditional value-at-risk*, denoted $\text{CVaR}_\alpha$. $\text{VaR}_\alpha$ corresponds to the $(1 - \alpha)$-quantile of the distribution. In other words, a $(1 - \alpha)$ fraction of the traces will have a loss smaller than the $\text{VaR}_\alpha$. To formally define it, let $Z_f \sim \mathcal{D}_f(T)$ be a random variable over the traces of the agent $f$. Then we can express

$$\text{VaR}_\alpha[L(Z_f)] := \text{quantile}(L(Z_f), 1 - \alpha) = \inf \left\{ q \in \mathbb{R} : \mathbf{Pr}[L(Z_f) \leq q] \geq 1 - \alpha \right\}. \quad (1)$$

On the other hand, $\text{CVaR}_\alpha$, a coherent risk measure Artzner et al. (1999), is more sensitive to the tail loss distribution and is defined as the expected tail loss:

$$\text{CVaR}_\alpha[L(Z_f)] := \frac{1}{\alpha} \int_0^\alpha \text{VaR}_\gamma[L(Z_f)] d\gamma = \mathbb{E}[L(Z_f) \mid L(Z_f) \geq \text{VaR}_\alpha[L(Z_f)]]. \quad (2)$$

We now shift our attention to compositions of agents. Agents can be composed sequentially to complete long-horizon objectives. A composition of agents sequentially transforms inputs in which the output of one agent becomes the input for the next agent. Given a long-horizon objective, we model all feasible agent compositions that achieve the objective as a directed acyclic graph called an *agent graph*.

**Definition** (Agent graph). *An agent graph $G$ is a tuple $(V, E, X, T, F, L, s, t, \mathcal{D}_s)$ with the following components:*

1. *A set of vertices $V$ and edges $E \subseteq V \times V$ that form a directed acyclic graph.*

2. *$X$ associates a domain $X_v$ to each vertex $v \in V$.*

3. *To each edge $e = (u, v) \in E$, $T$ associates a trace set $T_e$, $F$ associates an agent $f_e : X_u \to \mathfrak{D}(T_e \times X_v)$, and $L$ associates a loss function $L_e : T_e \to \mathbb{R}$.*

4. *$s \in V$ denotes the source vertex and $\mathcal{D}_s$ is the initial input distribution over the domain $X_s$.*

5. *$t \in V$ denotes the terminal vertex and $X_t$ represents the output domain of the long-horizon objective.*

Consider an agent graph $G$. Let $\mathcal{P} \subseteq V^*$ be the set of directed paths from $s$ to $t$. Consider $p = v_1 \xrightarrow{e_1} \dots \xrightarrow{e_m} v_{m+1} \in \mathcal{P}$ with $v_1 = s$ and $v_{m+1} = t$, and $e_1, \dots, e_m$ the directed edges along the path.

---

**Algorithm** BucketedVaR : Efficient algorithm for the RMAG-VaR objective

---

**Require:** $G = (V, E, X, T, F, L, s, t, \mathcal{D}_s)$: agent graph; $d \in \mathbb{N}_{>0}$: number of buckets; $n$: sample size; risk budget $\alpha \in (0, 1)$
1: $B \leftarrow \{0, \frac{\alpha}{d}, 2\frac{\alpha}{d}, \ldots, \alpha\}$ ▷ *Set of risk level buckets*
2: $\text{VaR} : (V \times B \rightarrow \mathbb{R}) \leftarrow$ init empty map ▷ *Map to store* VaR *estimates*
3: $\text{best\_paths} : (V \times B \rightarrow V^*) \leftarrow$ init empty map ▷ *Map to store best partial paths*
4: $\text{best\_samples} : (V \times B \rightarrow (X_v)^n) \leftarrow$ init empty map
5: ▷ *Map to store samples along best partial path*
6: $\text{VaR}[s, \_] \leftarrow -\infty; \text{best\_paths}[s, \_] \leftarrow [0, 0]$ ▷ *Base case: no risk at source*
7: $\text{best\_samples}[s, \_] \leftarrow [x_1, \ldots, x_n]$ where $x_1, \ldots, x_n \sim \mathcal{D}_s$ ▷ *Draw input samples*
8: $O \leftarrow$ topological ordering on $V$ excluding $s$
9: **for** vertex $v$ in the order $O$ and $\bar{\alpha} \in B$ **do**
10:     $\text{pred} \leftarrow$ predecessor vertices of $v$
11:     ▷ *All predecessors have been processed due to topological order*
12:     $\text{VaR}[v, \bar{\alpha}] \leftarrow \infty$
13:     **for** $v' \in \text{pred}$ and $\alpha' \in B_{\leq \bar{\alpha}}$ **do** ▷ $B_{\leq \bar{\alpha}} := \{q \in B : q \leq \bar{\alpha}\}$
14:         $f \leftarrow f_{(v, v')}; L \leftarrow L_{(v, v')}$
15:         $\{x_1, \ldots, x_n\} \leftarrow \text{best\_samples}[v', \alpha']$
16:         $\{(t_1, y_1), \ldots, (t_n, y_n)\} \leftarrow \{(t_1, y_1) \sim f(x_1), \ldots, (t_n, y_n) \sim f(x_n)\}$
17:         ▷ *Sample trace-output pairs along edge*
18:         $\text{losses} \leftarrow \{L(t_1), \ldots, L(t_n)\}$
19:         $\text{edgeVaR} \leftarrow \text{quantile}(\text{losses}, 1 - (\bar{\alpha} - \alpha'))$ ▷ *Empirical* $(1 - (\bar{\alpha} - \alpha'))$-*quantile*
20:         $\text{pathVaR} \leftarrow \max(\text{VaR}[v', \alpha'], \text{edgeVaR})$ ▷ *VaR along the path*
21:         **if** $\text{pathVaR} < \text{VaR}[v, \bar{\alpha}]$ **then**
22:             $\text{VaR}[v, \bar{\alpha}] \leftarrow \text{pathVaR}$ ▷ *Update best estimate*
23:             $\text{best\_paths}[v, \bar{\alpha}] \leftarrow \text{append}(\text{best\_paths}[v', \alpha'], v)$
24:             $\text{best\_samples}[v, \bar{\alpha}] \leftarrow \{y_1, \ldots, y_n\}$
25:         **end if**
26:     **end for**
27: **end for**
28: **return** $\text{VaR}[t, \alpha], \text{best\_paths}[t, \alpha]$ ▷ *Path with minimum VaR estimate*

---

Equipped with the initial input distribution $\mathcal{D}_s$, the sequence of agents $(f_{e_1}, \ldots, f_{e_m})$ induces a joint distribution $\mathcal{D}(T_{e_1} \times \cdots \times T_{e_m})$ which we denote $\mathcal{D}_p$. We can sample $(t_1, \ldots, t_m) \sim \mathcal{D}_p$ by sampling $x_1 \sim X_s$, followed by $(t_1, x_2) \sim f_{e_1}(x_1)$, $(t_2, x_3) \sim f_{e_2}(x_2)$, until $(t_m, x_{m+1}) \sim f_{e_m}(x_m)$. We refer to a sample from $\mathcal{D}_p$ as a *sequence of traces* and define the random variable $Z_p \sim \mathcal{D}_p$.

We also define a composed loss function along the path $L_p : T_{e_1} \times \cdots \times T_{e_m} \rightarrow \mathbb{R}$ as

$$L_p(t_1, \ldots, t_m) := \max_i \{L_{e_i}(t_i)\} \tag{3}$$

which is the maximum of losses incurred by the sequence of trajectories $(t_1, \ldots, t_m)$.

We now have all the ingredients to state our risk minimization objective over an agent graph (RMAG).

**Objective** (RMAG). *Let* $G = (V, E, X, T, F, L, s, t, \mathcal{D}_s)$ *be an agent graph and let* $\mathcal{P} \subseteq V^*$ *be the set of directed paths from* $s$ *to* $t$. *For a given risk level* $\alpha \in (0, 1)$ *and risk measure* $\rho \in \{\text{VaR}_\alpha, \text{CVaR}_\alpha\}$ *we define the risk minimization objective as the following optimization problem*

$$\underset{p \in \mathcal{P}}{\arg\min} \, \rho[L_p(Z_p)]. \tag{4}$$

Rephrasing in words, the optimization objective corresponds to finding a path from $s$ to $t$ in the agent graph $G$ that minimizes the risk measure of the *maximum* of the losses incurred by the agents along the path.

## 3 FINDING THE PATH MINIMIZING RISK

We first consider the RMAG objective with the VaR risk measure and then show we can recover the CVaR through our procedure. A simple procedure to find the optimal path is to estimate the

$\mathrm{VaR}_\alpha$ along each path and pick the path that minimizes it. We can estimate the $\mathrm{VaR}_\alpha$ by computing the empirical $(1 - \alpha)$-quantile on samples drawn from the loss along a path $L_p(Z_p)$. While this is asymptotically optimal, it can be inefficient when the graph has an exponential number of paths as a function of the number of vertices.

To circumvent this, we present an efficient algorithm that only scales polynomially in the number of vertices in the agent graph and is also asymptotically near-optimal under certain assumptions. BucketedVaR is a dynamic programming algorithm that processes the vertices in the agent graph in the topological order and incrementally builds a near-optimal path.

To understand how it works, we first present how we can estimate the value-at-risk of the loss distribution of a single path incrementally. To this end, consider a path in an agent graph $p = v_1 \xrightarrow{e_1} \ldots \xrightarrow{e_m} v_{m+1}$ with $v_1$ being the source vertex and $v_{m+1}$ the target vertex. Let $(Z_1, \ldots, Z_m) \sim \mathcal{D}(T_{e_1} \times \cdots \times T_{e_m})$ be a sequence of random variables representing the sequences of traces of the agents along the path. Then define $R_1, \ldots, R_m$ to be real-valued random variables representing the losses incurred by the agents along each edge. Formally, we can write $R_1 := L_{e_1}(Z_1), \ldots, R_m := L_{e_m}(Z_m)$. If $L_p$ is the composed loss function along the path as defined in Equation (3), we can rewrite $L_p(Z)$ as $\max(R_1, \ldots, R_m)$. A key observation that enables the incremental estimation of $\mathrm{VaR}_\alpha[L_p(Z)]$ (or equivalently, the $(1 - \alpha)$-quantile of $L_p(Z)$) is that we can estimate $\mathrm{VaR}_{\alpha_i}[R_i]$ of each loss variable individually with $\alpha_i$ chosen such that $\sum \alpha_i = \alpha$. Following this, we can apply the union bound to obtain

$$\mathrm{VaR}_\alpha[L_p(Z)] = \mathrm{VaR}_\alpha[\max(R_1, \ldots, R_m)] \leq \max_i\{\mathrm{VaR}_{\alpha_i}[R_i]\}. \tag{5}$$

This bound is justified in the proof of Theorem 1. This method allows us to estimate the value-at-risk of each edge loss variable independently and then combine the estimates at the end by computing their maximum. However, we need to optimize the allocation of the risk budgets $\alpha_i$ along each edge. Intuitively, allocating a higher risk budget to higher edge losses would result in a lower value-at-risk for that edge since it would correspond to a lower quantile.

BucketedVaR optimizes this by searching for risk budget allocations in a discretized set of *buckets* $B = \{0, \frac{\alpha}{d}, 2\frac{\alpha}{d}, \ldots, \alpha\}$ for a given discretization factor $d \in \mathbb{N}_{>0}$. At each vertex $v$ and corresponding to each bucket $\bar{\alpha} \in B$, the algorithm inductively builds an optimal path that minimizes the upper bound on the value-at-risk along the partial path at a risk budget of $\bar{\alpha}$. It achieves this by considering each predecessor $v'$ of $v$ and each allocation of budget $\alpha' \leq \bar{\alpha}$ until $v'$, for which we have inductively constructed the optimal path, and then estimates the value-at-risk by allocating a budget of $\bar{\alpha} - \alpha'$ to the loss along the edge $v' \to v$. The optimal path for $(v, \bar{\alpha})$ is then obtained by extending the optimal path to $(v', \alpha')$ that minimizes the value-at-risk estimate. A detailed proof of soundness and run-time analysis of the algorithm can be found in the proof of Theorem 1

**Theorem 1.** *Consider $G = (V, E, X, T, F, L, s, t, \mathcal{D}_s)$ an agent graph, $d \in \mathbb{N}_{>0}$ the number of buckets, $n \in \mathbb{N}_{>0}$ the sample size, and $\alpha \in (0, 1)$ the risk budget. Then, let $q \in \mathbb{R}$ and $p \in V^*$ be the value-at-risk estimate and path returned by $\mathrm{BucketedVaR}(G, d, n, \alpha)$. Then, for all $\delta \in (0, 1)$, with probability at least $1 - \delta$,*

$$q \geq \mathrm{quantile}(L_p(Z_p), 1 - \alpha - \gamma), \text{ with } \gamma = |V|\sqrt{\frac{1}{2n} \ln\left(\frac{2(d+1)^2|V|^2}{\delta}\right)} \tag{6}$$

*Furthermore, the time complexity of $\mathrm{BucketedVaR}(G, d, n, \alpha)$ is $O(n(d + 1)^2|V|^2)$ assuming that sampling from an agents' output distributions and the initial distribution $\mathcal{D}_s$ incur constant costs.*

*Proof sketch.* Observe that since $\gamma \to 0$ as $n \to \infty$, the theorem tells us that the estimated value-at-risk $q$ is at least as large as the true value-at-risk of the loss distribution of the path $p$, which corresponds to $\mathrm{quantile}(L_p(Z_p), 1 - \alpha)$. We can show this using a union bound argument for the budget allocations found by the algorithm along with the fact that the empirical CDF converges uniformly to the true CDF by the DKW inequality (Dvoretzky et al., 1956; Massart, 1990). The time complexity is easily deduced by looking at the pseudocode of BucketedVaR. See Appendix B for the complete proof. □

Theorem 1 only justifies the validity of the union bounding method for estimating the value-at-risk but we have not discussed the optimality of BucketedVaR. Here, we can only hope to achieve tight

estimates of the value-at-risk when the loss variables along every path are independent or are only loosely correlated. Under these independence assumptions, Theorem 2 tells us that asymptotically, the quantile estimated by the BucketedVaR algorithm is at most the $(1 - \alpha + \alpha^2/2)$-quantile of the losses along the *optimal* path. In other words, the path found by our algorithm is suboptimal by an additive factor of $\alpha^2/2$ with respect to the quantiles of the optimal path loss.

**Theorem 2.** *Let $G, d, n$, and $\alpha$ be defined as in Theorem 1. Further, assume that the losses along each edge of the agent graph are given by independent real-valued random variables $R_e$ for each $e \in E$ such that for all directed paths $p$, we can write $Z_p = (R_{e_1}, \ldots, R_{e_m})$. Now, let $(q, p) =$ BucketedVaR$(G, d, n, \alpha)$ and let $p^*$ be the path optimizing the* RMAG *objective. Then, for all $\delta \in (0, 1)$, with probability at least $1 - \delta$,*

$$q \underset{n, d \to \infty}{\leq} \text{quantile}\left( L_{p^*}\left( Z_{p^*} \right), 1 - \alpha + \frac{\alpha^2}{2} \right) \tag{7}$$

*Proof sketch.* This is again a consequence of the uniform convergence of the empirical CDF along with the fact that the independence of the loss variables allows us to deduce an upper bound. See Appendix B for the complete proof. □

Equipped with the BucketedVaR algorithm to approximate $\text{VaR}_\alpha$ over the agent graph, we show how we can also approximate the conditional value-at-risk $\text{CVaR}_\alpha$ using the quantities computed by the algorithm. From the expression for $\text{CVaR}_\alpha$ in Equation (2), it is easy to see that we can approximate the integral as

$$\text{CVaR}_\alpha[L_p(Z)] \approx \frac{1}{d} \sum_{k=1}^{d} \text{VaR}_{k\frac{\alpha}{d}}[L_p(Z)]. \tag{8}$$

But notice that the BucketedVaR algorithm has already optimized values of $\text{VaR}_{k\frac{\alpha}{d}}[L_p(Z)]$ over the agent graph for $1 \leq k \leq d$ and so it suffices to take the average of these values. Finally, we observe that increasing the sample size and the number of buckets also improves the approximation of $\text{CVaR}_\alpha$.

## 4 EXPERIMENTAL EVALUATION

### 4.1 BENCHMARKS

We implement the BucketedVaR algorithm for risk minimization and evaluate its performance on a suite of discrete and continuous control benchmarks that require composing agents trained with reinforcement learning (RL) to complete long-horizon objectives. In all the benchmarks that we consider, the agent graphs $G = (V, E, X, T, F, L, s, t, \mathcal{D}_s)$ share the following structure: the domain associated with each vertex $X_v$ is the state space of the control environment $\Sigma$ and the agents along each edge are control policies $\pi_e : \Sigma \to \mathfrak{D}(\Sigma^* \times \Sigma)$ that take an initial state and produce a distribution over trajectories and final states. These final states then become the initial states for the next policy. Lastly, $\mathcal{D}_s$ represents the distribution over the initial states of the environment.

In the first set of benchmarks, the objective is to *maximize safety during navigation and manipulation*. We adapt the 16-Rooms and Fetch environments from Jothimurugan et al. (2021) in which the long-horizon objective is specified as a *task graph*. In a task graph, each edge represents a *reach-avoid* subtask: reaching a target region while avoiding a set of dangerous states. To adapt this into an agent graph, we define loss functions for each edge subtask as the negative of the cumulative reward function which captures the reach-avoid specification. Additionally, we adapt the 16-Rooms environment to implement a version of the drone navigation task from Example 1. Secondly, we implement the BoxRelay benchmark using the Miniworld (Chevalier-Boisvert et al., 2023) framework in which we seek to *minimize resource consumption*. The objective is to move the player around while picking up boxes and dropping them at different locations. We add the constraint that the player has limited battery power but gets recharged each time a box is dropped off at a target. Importantly, since the initial positions of the player, the boxes, and the final target are all stochastic, this is a non-trivial optimization problem. To model the BoxRelay task as an agent graph, we define loss functions that evaluate to the number of time steps that a box is carried before being put back down and thus quantifies resource consumption. Complete benchmark descriptions are included in Appendix C.

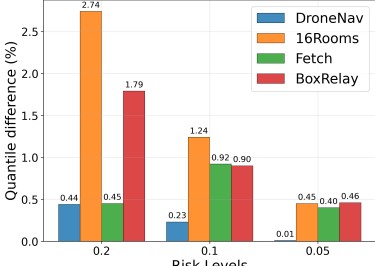 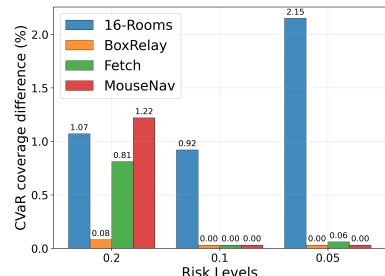

Figure 2: (a) Absolute quantile difference in percentage between the quantile computed by the BucketedVaR algorithm and the desired quantile. (b) Absolute quantile difference in percentage between the approximate $CVaR_\alpha$ and that estimated by the baseline. The empirical quantiles are computed on a fresh set of $10^4$ samples.

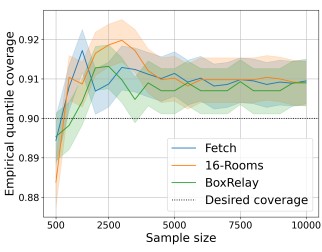 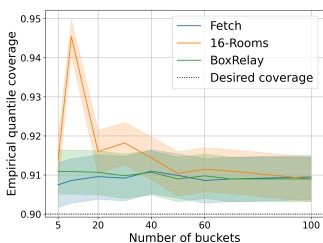 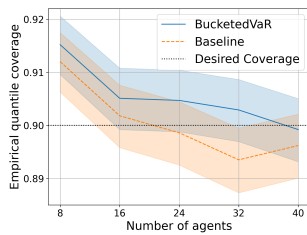

(a) Varying sample size      (b) Varying number of buckets      (c) Varying number of agents in path

Figure 3: Empirical quantiles of the $VaR_{0.1}$ estimates computed by BucketedVaR with varying parameters. $95\%$ Clopper-Pearson CIs for the empirical quantiles computed on $10^4$ samples are also plotted.

## 4.2 Experimental results

Equipped with these benchmark environments, we conduct three sets of experiments to evaluate the effectiveness of the approximation obtained from the BucketedVaR algorithm. We measure the accuracy of the approximation by estimating the empirical quantile on $10^4$ fresh samples of losses along the optimal path. Specifically, if $c \in \mathbb{R}$ is the approximate $VaR_\alpha$ returned by the BucketedVaR algorithm, we draw samples of losses along the optimal path and compute the fraction of these which are below $c$ giving us the *empirical quantile* of $c$. We say that the approximation is tight if the empirical quantile is close to the desired quantile $1 - \alpha$. We also use empirical quantiles to compare approximations of $CVaR_\alpha$.

**Experiment 1: Comparison with optimal baseline algorithm.** We define a baseline algorithm that explicitly esimates $VaR_\alpha$ and $CVaR_\alpha$ at the desired risk level $\alpha$ along each path of the agent graph. While this algorithm is asymptotically optimal, it is inefficient as discussed in the previous section. The number of samples for both algorithms is set to $10^4$ and the number of buckets is chosen based on the number of agents to allow sufficient granularity in the allocation of the risk budget. For example, in the 16-Rooms benchmark since there are 8 agents along each path, we choose a higher number of buckets.

We find that BucketedVaR succeeds in finding the same optimal path as the baseline algorithm across all the benchmarks along with tight estimates of both $VaR_\alpha$ and $CVaR_\alpha$. Figure 2 shows the absolute difference in percentage points of the quantile estimates obtained by the BucketedVaR algorithm and the baseline quantiles. We remark that the difference is never more than a couple percentage points across all benchmarks and risk levels. It also finds non-trivial allocations of the risk budget for the agents: in the 16-Rooms benchmark, the budget allocations for the 8 agents along the path to estimate $VaR_{0.1}$ are $16\bar{\alpha}, 0\bar{\alpha}, 10\bar{\alpha}, 23\bar{\alpha}, 19\bar{\alpha}, 11\bar{\alpha}, 7\bar{\alpha}$, and $14\bar{\alpha}$ for $\bar{\alpha} = 0.1/100$.

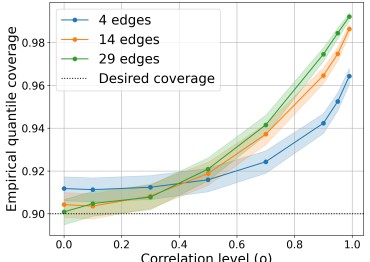 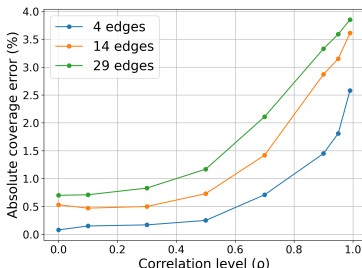

Figure 4: (a) Empirical VaR$_{0.1}$ coverage produced by BucketedVaR algorithm with respect to increasing correlation and path length. (b) Absolute coverage error (in percent) between CVaR$_{0.1}$ estimate produced by BucketedVaR and the baseline estimate. This is also plotted against increasing correlation and path length.

**Experiment 2: Increasing sample size and number of buckets.** Increasing the sample size improves the approximation because we have more accurate quantile estimates and similarly a higher number of buckets results in more granular risk budget allocations. This is confirmed by the two experiments we run: in the first one, plotted in Figure 3a, we maintain the number of buckets at 100 and vary the number of samples from 500 to $10^4$ with $\alpha = 0.1$. The empirical quantile quickly stabilizes around 0.91 across benchmarks. The second experiment maintains the sample size at $10^4$ and varies the number of buckets from 5 to 100 with $\alpha = 0.1$. Figure 3b again shows a similar trend. We note that for the 16-Rooms benchmark, a low number of buckets gives bad VaR estimates since the number of agents along each path is higher.

**Experiment 3: Scaling number of agents.** From our theoretical result, we expect BucketedVaR to continue producing tight estimations of the VaR when we increase the number of agents along the path, provided that the condition on the independence of losses holds and the sample size is large enough. We test this hypothesis by simulating a long path of agents using the 16-Rooms benchmark. We consider the optimal path in this benchmark which has a sequence of 8 agents and construct a new agent graph with $8k$ agents for $k \in \{1, 2, 3, 4, 5\}$. We can sample a sequence of trajectories from this path by taking $k$ trajectories of the original 8 agents. The losses are then computed using the original loss functions. Since these are independently drawn trajectories, the assumption of independence holds. We estimate the VaR$_\alpha$ of this path of agents using BucketedVaR for $10^4$ samples, $\alpha = 0.1$, and 100 buckets. The results are plotted in Figure 3c in which we observe that the approximation continues to remain tight with an increasing number of agents.

**Experiment 4: Robustness to correlated losses.** To test the quality of the union bound approximations made by the BucketedVaR algorithm when losses are not independent, we setup an artificial agent graph which consists of a path of varying length along with loss functions that sample from correlated Gaussian distributions (complete description in Appendix C.3). This allows us to study the tightness of the approximation with increasing correlation ($\rho$) and path length. The results are plotted in Figure 4. While the approximation breaks down when we have complete correlation ($\rho = 1$), it is interesting to note that the algorithm continues to produce reasonable approximations upto a correlation level of $\rho = 0.5$.

## 5 LIMITATIONS

A limitation of our algorithm is that we require the losses of agents to be independent to obtain the theoretical guarantee regarding the tightness of the value-at-risk estimates. Empirically, however, we observe that it continues to produce tight estimates even without any formal independence guarantees which indicates that we may be able to relax this assumption. Regardless, we emphasize that this independence assumption holds when the losses are metrics of the agent behavior rather than task performance which is the case in all our benchmarks and examples. Another limitation is that we may not always have well defined loss functions like in the case of an LLM agent and we may have to rely on LLM-as-a-judge methods or human annotations which may be expensive and noisy. Lastly, the sample complexity of our method can be high to obtain accurate risk measure estimates.

However, if our only objective is to find the optimal path without computing risk measures, then online optimization using a multi-armed bandits style approach could be explored.

## 6 CONCLUSION AND FUTURE DIRECTIONS

In summary, we introduced a framework for choosing optimal agent compositions to minimize the value-at-risk of losses that encode violations of safety, fairness, and privacy requirements. Our dynamic programming approach efficiently allocates the risk budget across agents, supporting tight approximations. Empirical results on compositional RL benchmarks confirm its effectiveness in identifying optimal paths and quantifying tail behaviors, paving the way for more reliable agentic workflows.

A direction for the future is to incorporate the BucketedVaR algorithm in existing agentic frameworks. It is also possible to speed up the algorithm through a GPU implementation since the computation of the intermediate VaR estimates is highly parallelizable. While we present an efficient algorithm to minimize the value-at-risk in agent graphs, it is also an interesting future direction to extend our work to the estimation of other popular risk measures that are included in the class of coherent risk measures (Artzner et al., 1999). Yet another interesting direction is to explore whether we can develop online multi-armed bandit style algorithms to minimize risk. Such an algorithm would have to pick a composition of agents at each turn and minimize long-term regret with respect to the optimal composition. This can reduce sample complexity by directly converging towards the optimal composition circumventing the risk measure computation. Recent works in risk-aware bandits (Tan et al., 2022) and combinatorial bandits (Ayyagari and Dukkipati, 2021) seem to indicate that this is feasible.

## 7 REPRODUCIBILITY STATEMENT

The source code along with detailed instructions to reproduce results and plots have been included in the supplementary materials. Further benchmark and experimental details are also included in Appendix C.

### ACKNOWLEDGEMENTS

This research was partially supported by the NSF Award SLES 2331783.

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

## A    USE OF LARGE LANGUAGE MODELS (LLMS)

LLMs were used as code assistants in writing and debugging parts of the codebase. They were also used to polish the writing.

## B    PROOFS OF CORRECTNESS, EFFICIENCY, AND OPTIMALITY OF THE BUCKETEDVAR ALGORITHM

*Proof of Theorem 1.* We first establish the correctness of the algorithm by showing that, with high probability, the value-at-risk estimate $q$ returned by the BucketedVaR algorithm is at least as large as the $(1 - \alpha - \gamma)$-quantile of the loss distribution along the returned path $p$.

Let $p = v_1 \to v_2 \to \cdots \to v_k$ be the path returned by the algorithm, where $v_1 = s$ and $v_k = t$. For $i \in \{1, \ldots, k - 1\}$, let $e_i = (v_i, v_{i+1})$ be the edges along this path, and let $\alpha_i$ be the risk budget allocated to edge $e_i$ such that $\sum_{i=1}^{k-1} \alpha_i = \alpha$. These allocations correspond to the buckets selected by the algorithm.

For each edge $e_i$, let $R_i = L_{e_i}(Z_i)$ be the random variable representing the loss along that edge. The loss along the entire path is given by $L_p(Z_p) = \max(R_1, \ldots, R_{k-1})$.

By the union bound property of probabilities, we have

$$\Pr\left[\max(R_1, \ldots, R_{k-1}) > q\right] \leq \sum_{i=1}^{k-1} \Pr[R_i > q] \tag{9}$$

$$= \sum_{i=1}^{k-1} \alpha_i \tag{10}$$

$$= \alpha \tag{11}$$

where we set $q$ such that $\Pr[R_i > q] = \alpha_i$.

This implies that $q \geq \text{VaR}_\alpha[L_p(Z_p)]$. However, in the algorithm, we estimate the $(1 - \alpha_i)$-quantile for each edge empirically using $n$ samples. By the Dvoretzky-Kiefer-Wolfowitz (DKW) inequality (Dvoretzky et al., 1956; Massart, 1990), with probability at least $1 - \delta'$, the empirical CDF $\hat{F}_i$ for each edge satisfies:

$$\sup_{x \in \mathbb{R}} |\hat{F}_i(x) - F_i(x)| \leq \sqrt{\frac{1}{2n} \ln \frac{2}{\delta'}} \tag{12}$$

where $F_i$ is the true CDF of $R_i$.

For our algorithm, we need this to hold for all the empirical CDFs that we estimate. By looking at the pseudocode, we remark that we make at most $|V|^2(d + 1)^2$ quantile estimations. So setting $\delta' = \frac{\delta}{(d+1)^2|V|^2}$ and applying the union bound, with probability at least $1 - \delta$, all empirical CDFs are within $\sqrt{\frac{1}{2n} \ln \frac{2(d+1)^2|V|^2}{\delta}}$ of their true CDFs.

Therefore, the error in quantile estimation for the path $p$ is at most

$$\gamma = (k-1)\sqrt{\frac{1}{2n} \ln\left(\frac{2(d+1)^2|V|^2}{\delta}\right)} \leq |V|\sqrt{\frac{1}{2n} \ln\left(\frac{2(d+1)^2|V|^2}{\delta}\right)}. \tag{13}$$

This implies that, with probability at least $1 - \delta$, the estimated value-at-risk $q$ is at least as large as the $(1 - \alpha - \gamma)$-quantile of $L_p(Z_p)$.

For the time complexity analysis, observe that the algorithm processes each vertex-bucket pair $(v, \bar{\alpha})$ at most once and there are at most $|V|(d + 1)$ such pairs. For each pair, it considers all its predecessors vertices $v'$ and bucket allocation $\alpha' \in B_{\leq \bar{\alpha}}$. There are again at most $|V|(d + 1)$ pairs. For each combination, it processes $n$ samples. Thus, the overall time complexity is $O(n(d + 1)^2|V|^2)$, assuming constant-time sampling operations. □

*Proof of Theorem 2.* Under the independence assumption of losses along edges, we can provide a tighter analysis of the optimality of the BucketedVaR algorithm.

Let $p^* = v_1^* \to v_2^* \to \cdots \to v_{k^*}^*$ be the optimal path minimizing the value-at-risk, where $v_1^* = s$ and $v_{k^*}^* = t$. For $i \in \{1, \ldots, k^* - 1\}$, let $e_i^* = (v_i^*, v_{i+1}^*)$ be the edges along this path.

Let $R_1^*, \ldots, R_{k^*-1}^*$ be the independent random variables representing the losses along the edges of the optimal path. The loss along the entire optimal path is given by $L_{p^*}(Z_{p^*}) = \max(R_1^*, \ldots, R_{k^*-1}^*)$.

For independent random variables, the CDF of the maximum is the product of the individual CDFs

$$F_{\max}(x) = \Pr[\max(R_1^*, \ldots, R_{k^*-1}^*) \leq x] = \prod_{i=1}^{k^*-1} \Pr[R_i^* \leq x] = \prod_{i=1}^{k^*-1} F_i(x) \qquad (14)$$

Now, suppose we optimally allocate the risk budget $\alpha$ among the edges of the optimal path, assigning $\alpha_i^*$ to edge $e_i^*$ such that $\sum_{i=1}^{k^*-1} \alpha_i^* = \alpha$.

The optimal allocation would ensure that the $(1 - \alpha_i^*)$-quantile is the same for all edges, which we denote as $q^*$. This means $F_i(q^*) = 1 - \alpha_i^*$ for all $i$. Since BucketedVaR searches over discretized buckets, as $d \to \infty$, it approaches this optimal allocation.

Under this optimal allocation, the CDF of the maximum at $q^*$ is

$$F_{\max}(q^*) = \prod_{i=1}^{k^*-1} F_i(q^*) \qquad (15)$$

$$= \prod_{i=1}^{k^*-1} (1 - \alpha_i^*) \qquad (16)$$

To upper bound this product, we use the inequality $(1 - x) \leq e^{-x}$ which holds for all $x \in [0, 1]$

$$\prod_{i=1}^{k^*-1} (1 - \alpha_i^*) \leq \prod_{i=1}^{k^*-1} e^{-\alpha_i^*} \qquad (17)$$

$$= e^{-\sum_{i=1}^{k^*-1} \alpha_i^*} \qquad (18)$$

$$= e^{-\alpha} \qquad (19)$$

For all $\alpha > 0$, we can use the Taylor expansion of $e^{-\alpha}$ around zero

$$e^{-\alpha} = 1 - \alpha + \frac{\alpha^2}{2!} - \frac{\alpha^3}{3!} + \frac{\alpha^4}{4!} - \ldots \qquad (20)$$

To establish a strict upper bound on $e^{-\alpha}$, we consider the function $h(\alpha) = 1 - \alpha + \frac{\alpha^2}{2} - e^{-\alpha}$ and show that $h(\alpha) > 0$ for all $\alpha > 0$.

Note that $h(0) = 0$ and $h'(\alpha) = -1 + \alpha + e^{-\alpha}$. At $\alpha = 0$, we have $h'(0) = 0$. Computing the second derivative, $h''(\alpha) = 1 - e^{-\alpha}$, we see that $h''(\alpha) > 0$ for all $\alpha > 0$ since $e^{-\alpha} < 1$ when $\alpha > 0$.

Since $h''(\alpha) > 0$ for all $\alpha > 0$, the function $h'(\alpha)$ is strictly increasing for $\alpha > 0$. Given that $h'(0) = 0$, this means $h'(\alpha) > 0$ for all $\alpha > 0$.

As $h'(\alpha) > 0$ for all $\alpha > 0$, the function $h(\alpha)$ is strictly increasing for $\alpha > 0$. Since $h(0) = 0$, we conclude that $h(\alpha) > 0$ for all $\alpha > 0$.

Therefore, we have established the strict inequality

$$e^{-\alpha} < 1 - \alpha + \frac{\alpha^2}{2} \quad \text{for all } \alpha > 0 \qquad (21)$$



Figure 5: 16-Rooms environment with obstacles to avoid in red. The light blue square in the bottom left corner is the initial room of the point mass. The first subgoal is the dark blue square, followed by green, pink, and brown. Image taken from Jothimurugan et al. (2021).

This gives us

$$F_{\max}(q^*) \leq 1 - \alpha + \frac{\alpha^2}{2} \tag{22}$$

which implies that

$$q^* \leq \text{quantile}(L_{p^*}(Z_{p^*}), 1 - \alpha + \frac{\alpha^2}{2}) \tag{23}$$

As $n \to \infty$, the empirical quantile estimates converge to the true quantiles, and as $d \to \infty$, the discretization error in budget allocation vanishes. Therefore, asymptotically, the value-at-risk estimate $q$ returned by BucketedVaR satisfies $q \leq \text{quantile}(L_{p^*}(Z_{p^*}), 1 - \alpha + \frac{\alpha^2}{2})$, which completes the proof. □

## C  BENCHMARK ENVIRONMENTS AND EXPERIMENTAL EVALUATION SETUPS

### C.1  MAXIMIZING SAFETY DURING NAVIGATION AND MANIPULATION

We test our algorithm on the 16-Rooms and Fetch environments from Jothimurugan et al. (2021) who present a compositional RL framework to complete long-horizon objectives in continuous environments. An objective is specified as a *task graph* which is the same as an agent graph. In the task graph, every edge represents a *reach-avoid* subtask: reaching a target region while avoiding a set of dangerous states. They design an algorithm that finds the path that maximizes the probability of completing all the reach-avoid subtasks along the path where the control policies along the edges are trained using RL. Each reach-avoid subtask specifies a real-valued reward function that takes in a trajectory of execution of the policy and evaluates to at least zero if the trajectory successfully completes the subtask. Intuitively, the reward captures a distance metric between trajectories and reach-avoid sets. The loss function in the agent graph is defined as the negative of the reward function.

The 16-Rooms environment consists of 16 rooms arranged in a $4 \times 4$ grid with doors between adjacent rooms. RL policies are trained to control a point mass to complete reach-avoid tasks. The agent graph has 13 vertices and 16 paths of length 8 each. The second Fetch environment (Plappert et al., 2018) is taken from Gymnasium-Robotics and involves controlling a 7-DoF robotic arm with a gripper attached at the end to pick up and move objects. We consider the task graph for the PickAndPlaceChoice objective that directs the gripper to move near the object, grip it, pick it up and move along one of two trajectories. The graph has 7 vertices and 2 paths of length 5. Additionally, we adapt the Rooms environment to implement a version of the drone navigation task from Example 1.

### C.1.1  16-ROOMS

**Environment and agent graph.**  This benchmark environment is borrowed from Jothimurugan et al. (2021). The state and action spaces are continuous and involves controlling a point mass to navigate between rooms while avoiding obstacles. A map of the environment is shown is shown in Figure 5. Jothimurugan et al. (2021) specify objectives in the form of task graphs where each vertex corresponds to a set of subgoal states and each edge represents a reach-avoid subtask of reaching the target vertex's subgoal states. The task graph that we consider corresponds to the specification $\phi_5$ from Jothimurugan et al. (2021) and is also visualized in Figure 5. It consists of 13 vertices arranged in a sequence of 4 diamonds with 4 vertices in each diamond like in Figure 1. The first diamond specifies starting from the light blue square and reaching the room marked with the dark blue square in Figure 5 using either the top path while avoiding the red obstacle or using the bottom path. Similarly, the next diamond specifies the two paths to reach the green square, followed by the pink, and brown squares. Each edge in the task graph is also associated with a real-valued reward function that evaluates to at least zero when a trajectory of the controller satisfies the reach-avoid subtask. This is function of the distance between a trajectory and the obstacles and between the trajectory and the subgoal region. So if the reward is positive then it means that the trajectory stays at least a certain distance away from the obstacles and reaches at most a certain distance to the goal.

Mapping a task graph to an agent graph is simple: to define the loss functions along each edge, we simply consider the negative of the reward function so that an upper bound on the loss corresponds to a lower bound on the reward. The agents along each edge are RL policies trained to complete the reach avoid subtasks. The policies are trained using the ARS algorithm (Mania et al., 2018) using the same recipe as described in Jothimurugan et al. (2021). Since the training of policies are dependent on the initial state distribution, we train the policies lazily when required by the BucketedVaR algorithm by approximating the initial state distribution from samples from the previous policy.

**Additional experimental observations.**  In Table 1, since the estimated VaR at all levels for the 16-Rooms benchmark is negative, it implies that with probability at least $1 - \alpha$, the reward along all the edges of the path are at least the positive estimated VaR, where $\alpha$ is the risk level.

### C.1.2  FETCH

The Fetch robotic arm environment (Plappert et al., 2018) from Gymnasium Robotics is a simulated robot arm environment with a gripper attached. The reach-avoid task graph is again taken from Jothimurugan et al. (2021) and has 8 vertices with two paths. It corresponds to the PickAndPlaceChoice specification and involves moving the gripper close to an object, gripping it, picking it up, and moving it along one of two trajectories. The policies are trained using TD3 (Fujimoto et al., 2018) with the hyperparameters specified by Jothimurugan et al. (2021).

### C.1.3  DRONENAV

As described previously, we adapt the 16-Rooms environment to implement the DroneNav environment from Example 1. Additionally, we adjust the sizes of the obstacles so that one of the paths has bigger obstacles than the other path.

**Additional experimental observations.**  This environment was designed as a counterexample—the algorithm from Jothimurugan et al. (2021) would be unable to pick a path in the graph since it only maximizes the success probability and in this case both are feasible to complete the reach-avoid tasks with probability 1. On the other hand, optimizing the RMAG objective would give us that the path with the smaller obstacles is safer.

### C.2  MINIMIZING RESOURCE CONSUMPTION

We use the Miniworld (Chevalier-Boisvert et al., 2023) framework to implement the BoxRelay environment of which the top-view map is shown in Figure 6a. Observations in this environment are given as first-person view RGB images (see Appendix C.2.1). The objective is to move the player

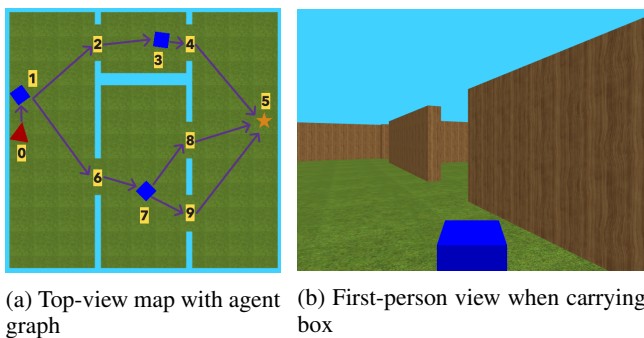

(a) Top-view map with agent graph

(b) First-person view when carrying box

Figure 6: BoxRelay environment

(red triangle) to the first box (marked 1) and pick it up. The player can then choose to go to either of the two boxes in the two middle rooms (3 or 7), place the initial box, pick up the next box, and finally drop it at the location marked in the right room with a star. Furthermore, after picking up the box at 7, the player can choose to exit through one of the two exits 8 or 9. We add the constraint that the player has limited battery power but gets recharged after dropping the first box and before picking up the second box. So we would like to minimize the amount of time that either box is held by the player. Importantly, since the initial positions of the player, the boxes, and the final target are all chosen uniformly at random within each room, this is a non-trivial optimization problem.

We can model the BoxRelay task as an agent graph as follows: the graph is as shown in Figure 6a with the numbers being the vertices. The edges represent the agents which are RL policies trained to navigate the player to pick up and place objects at the desired locations. The loss functions evaluate to the number of time steps that a box is carried before being put back down and thus quantifies the resource consumption. Here, the RMAG objective corresponds to minimizing the *maximum* time that any box is carried by the player.

### C.2.1 BOXRELAY

The environment and the agent graph details are presented in Section 4. The policies are trained using the PPO (Schulman et al., 2017) CNN-Policy algorithm from StableBaselines3 package with the default hyperparameters for 500,000 iterations for each edge policy. The initial state distribution was again estimated by taking rollouts of the previous edge policy. The loss functions in the agent graph are defined as follows where the vertices are numbered as shown in Figure 6a.

- $L_e(t) := -\infty$ for edges $e \in \{(0,1), (1,2), (1,6), (3,4), (7,8), (7,9)\}$ and for all trajectories $t$.

- The other edge loss functions for $e \in \{(2,3), (6,7), (4,5), (8,9), (9,5)\}$ return the number of time steps that the box was carried by the player by cumulating it with the number of time steps from the previous agent which also carried the box.

### C.3 CORRELATED GAUSSIAN NOISE EXPERIMENT

For experiment 4, to measure the robustness of our algorithm to correlations in the loss distributions, we create a synthetic agent graph which is simply a path with a varying number of edges $p > 0$. Let $C, X_1, \ldots, X_p \sim N(0,1)$ be i.i.d. Gaussian variables with zero mean and unit variance. Then for a given correlation level $\rho \in [0,1]$, we define the correlated edge loss variables $L_1, \ldots, L_p$ as

$$L_i := \rho C + \sqrt{1 - \rho^2} X_i. \tag{24}$$

Thus, under complete correlation $\rho = 1$, all the edge loss random variables represent the same variable $C$ and under zero correlation $\rho = 0$, they are independent.

Table 1: Comparison of the approximated VaR with the optimal baseline algorithm to compute VaR on $10^4$ samples. We prefer empirical quantiles that are close to $1 - \alpha$ where $\alpha$ is the risk level. 95% Clopper-Pearson CIs are specified for the empirical quantiles computed on a fresh set of $10^4$ samples.

| Benchmark | Risk level | Buckets | VaR estimate | Baseline estimate | VaR quantile (%) | Baseline quantile (%) |
|---|---|---|---|---|---|---|
| DroneNav | 0.2 | 5 | -0.339 | -0.339 | $79.56^{[78.75, 80.34]}$ | $79.56^{[78.75, 80.34]}$ |
| | 0.1 | 5 | -0.330 | -0.330 | $89.77^{[89.15, 90.35]}$ | $89.77^{[89.15, 90.35]}$ |
| | 0.05 | 5 | -0.328 | -0.328 | $95.01^{[94.56, 95.42]}$ | $95.01^{[94.56, 95.42]}$ |
| 16-Rooms | 0.2 | 100 | -0.0253 | -0.0277 | $82.74^{[81.98, 83.47]}$ | $81.26^{[80.48, 82.02]}$ |
| | 0.1 | 100 | -0.0135 | -0.0135 | $91.24^{[90.66, 91.78]}$ | $91.24^{[90.66, 91.78]}$ |
| | 0.05 | 100 | -0.0084 | -0.0084 | $95.45^{[95.02, 95.85]}$ | $95.45^{[95.02, 95.85]}$ |
| Fetch | 0.2 | 30 | 0.0314 | 0.0250 | $80.45^{[79.65, 81.22]}$ | $79.58^{[78.77, 80.36]}$ |
| | 0.1 | 30 | 0.3483 | 0.3393 | $90.92^{[90.33, 91.47]}$ | $90.53^{[89.93, 91.09]}$ |
| | 0.05 | 30 | 0.3914 | 0.3881 | $95.40^{[94.97, 95.80]}$ | $95.28^{[94.84, 95.68]}$ |
| BoxRelay | 0.2 | 50 | 151 | 148 | $81.79^{[81.01, 82.54]}$ | $80.56^{[79.77, 81.33]}$ |
| | 0.1 | 50 | 176 | 175 | $90.90^{[90.31, 91.45]}$ | $90.70^{[90.11, 91.26]}$ |
| | 0.05 | 50 | 210 | 208 | $95.46^{[95.03, 95.85]}$ | $95.34^{[94.90, 95.74]}$ |

Table 2: Comparison of the approximated CVaR with the optimal baseline algorithm on $10^4$ samples. Empirical coverages are computed on a fresh set of $10^4$ samples.

| Benchmark | Risk level | Buckets | CVaR estimate | Baseline estimate | CVaR quantile (%) | Baseline quantile (%) |
|---|---|---|---|---|---|---|
| DroneNav | 0.2 | 30 | 17.648 | 17.649 | 87.08 | 88.30 |
| | 0.1 | 30 | 17.652 | 17.652 | 94.72 | 94.72 |
| | 0.05 | 30 | 17.654 | 17.654 | 96.99 | 96.99 |
| 16-Rooms | 0.2 | 100 | -0.0134 | -0.0139 | 90.82 | 89.75 |
| | 0.1 | 100 | -0.0077 | -0.0073 | 95.26 | 94.34 |
| | 0.05 | 100 | -0.0051 | -0.0045 | 96.55 | 98.70 |
| Fetch | 0.2 | 30 | 0.359 | 0.317 | 88.96 | 88.15 |
| | 0.1 | 30 | 0.411 | 0.409 | 96.74 | 96.74 |
| | 0.05 | 30 | 0.451 | 0.453 | 98.19 | 98.25 |
| BoxRelay | 0.2 | 50 | 180.54 | 179.86 | 93.57 | 93.49 |
| | 0.1 | 50 | 210.96 | 210.61 | 96.47 | 96.47 |
| | 0.05 | 50 | 246.30 | 246.27 | 98.24 | 98.24 |

## C.4 EXPERIMENT COMPUTATIONAL REQUIREMENTS

All experiments were completed on one computer with an Intel Xeon 6248 CPU and one NVIDIA GeForce RTX 2080 GPU.

## D EXPERIMENTAL RESULTS

Complete evaluation results of the BucketedVaR algorithm along with comparison with the baseline optimal algorithms to compute VaR and CVaR are given in Table 1.

