# OpenReview forum: "Risk-Sensitive Agent Compositions"
_ICLR.cc/2026/Conference — ICLR 2026 Poster_

### Official Review · Reviewer_ZHyK · 2025-10-21

**Soundness:** 3
**Presentation:** 3
**Contribution:** 3
**Rating:** 6
**Confidence:** 3

**Summary:**

The paper studies risk-sensitive agent composition in multi-agent workflows. It models the system as a directed acyclic graph (DAG), where each edge represents an agent and each path corresponds to a feasible composition. The goal is to minimize the Value-at-Risk (VaR) of losses that capture safety, fairness, or privacy violations, focusing on tail risks. To do this, the authors propose a dynamic programming (DP)–style algorithm, BucketedVaR, that uses a union bound and a discretized risk budget to estimate an upper bound on the VaR. They prove asymptotic near-optimality under independence assumptions, and experiments on compositional RL benchmarks show the method can recover the optimal paths with accurate VaR estimates.

**Strengths:**

1. The paper presents a technically sound and mathematically coherent formulation of risk minimization for agent compositions.

2. The theoretical development is detailed and carefully reasoned, with clear proofs supporting the approximation guarantees.

3. Overall, the work provides both a mathematical framework and an algorithmic contribution that meaningfully advance the study of risk-aware agent systems.

**Weaknesses:**

1. The paper’s presentation is quite dense, and the abstract and introduction do not clearly communicate the main ideas, which makes it difficult to identify the core contribution without considerable effort.

2. The approach also depends on several strong assumptions, such as the DAG workflow structure and the independence of agent losses, which limit its generality and practical relevance.

3. Finally, the work remains mostly theoretical, and it is not clear how the proposed framework could be applied or integrated into real-world multi-agent systems.

**Questions:**

1. The framework assumes that the overall task can be represented as a DAG of agents. In many real-world scenarios, agent interactions may not follow a strict directional or decomposable structure. How restrictive is this assumption, and could the proposed approach be extended beyond DAG workflows?

2. The theoretical analysis seems to rely on several independence and directionality assumptions, as well as implicitly fixed outputs between connected agents. Could the authors clarify which of these are essential for the algorithm’s validity and which could be relaxed?

3. The loss function is defined only over the trace T, while the output Y appears not to influence the risk. In realistic settings, the output could affect downstream traces. How would the framework handle such dependencies?

4. In Figure 4b, the 16-Rooms benchmark exhibits a noticeable spike when using 5–20 buckets. Could the authors explain this behavior?

5. While the theoretical and algorithmic contributions are clear, the practical applicability of the framework remains uncertain. The paper would benefit from a discussion of potential real-world scenarios where the proposed method could be directly used or integrated into existing agentic systems.

I am not primarily working on the formal side of risk-sensitive optimization, but these questions reflect my current understanding of the framework and its assumptions.

---

> ### Author Response · Authors · 2025-11-19
>
> We thank the reviewer for their detailed comments and suggestions. In this first revision, we have made two additions:
>
> 1. **Computation of CVaR**: As most of the reviewers pointed out, conditional value-at-risk, CVaR, is a more useful risk measure as it is sensitive to the tail risk distribution and is a coherent risk measure. We show how we can obtain an approximation of CVaR directly from the computations made by our algorithm and also demonstrate empirically that these approximations are tight on our benchmarks.
>    Since the conditional value-at-risk is the expected tail quantile, it can be approximated by the average of the tail quantiles already computed by our algorithm. Figure 2(b) shows that this approximated CVaR is very close to the baseline CVaR computed by explicit path enumeration.
> 2. **Robustness to correlations in losses**: We implement a new benchmark where we can control the amount of correlation between edge losses and show that our algorithm is robust to a reasonable amount of correlation (up to $\\rho \= 0.5$).
>    This benchmark consists of a synthetic agent graph with a path of varying length where the losses along all the edges are distributed according to correlated Gaussians. We conduct experiments on this benchmark by using our algorithm to approximate the VaR and CVaR while varying the correlation level and path length. As seen in Figure 4, the approximation breaks down under complete correlation but is still robust to some correlation.
>
> Additions in the text are marked by a red font.
>
> Here are detailed responses to the the questions raised:
>
> 1. **Correlated/dependent losses**: The additional experiment that we have incorporated demonstrates that our algorithm is robust to some amount of correlation. Beyond this, we believe it is hard to do better without explicit path enumeration in the black-box setting. Additionally, we remark that the computational trace of an agent can include its output. However, in practice, we observe that agent losses are only weakly correlated when the losses encode objectives that are orthogonal to task success. This was the main motivation behind defining the loss as a function of the trace and excluding the output.
> 2. **Parallel agent compositions**: The reviewer points out the interesting case where agents are composed in parallel instead of sequentially. While this could be addressed by treating parallel agents as a single agent, it also motivates the harder problem of analysing their behaviors independently and then combining their guarantees.
> 3. **Significance**: The primary novel contribution of this work is certainly of theoretical nature in the form of an efficient sampling-based algorithm to estimate tail quantiles over DAGs. As demonstrated in the examples in the introduction, we believe there is ample scope to apply the algorithm for LLM agents and robotic task planning. For instance, a company deploying agentic software as a service can make use of simulated user scenarios to obtain samples and apply our algorithm. Alternatively, they could also incrementally optimize the agent composition that minimizes the risk as they collect more user data.
> 4. **Spike with 5-20 buckets in 16Rooms benchmark**: Since each path in the 16Rooms agent graph is of length 8, the number of buckets 5-20 is insufficient to properly approximate the VaR and this is a discretization error. While the jump appears significant with respect to the other curves, it only represents a 3% approximation error jump.

---

> > ### Comment · Reviewer_ZHyK · 2025-11-19
> >
> > I appreciate the authors’ response. However, after carefully reading it, I was only able to identify answers to Question 4 and partially to Question 5. Given this mismatch, I am not sure whether the authors intended this rebuttal for a different reviewer.

---

> > > ### Author Response · Authors · 2025-11-19
> > >
> > > Apologies for the lack of clarity. We tried to address the main weaknesses and questions but indeed there remain a few unaddressed remarks. Here are some more detailed responses:
> > >
> > > *1. The framework assumes that the overall task can be represented as a DAG of agents. In many real-world scenarios, agent interactions may not follow a strict directional or decomposable structure. How restrictive is this assumption, and could the proposed approach be extended beyond DAG workflows?*
> > >
> > > DAGs capture many real-world task workflows and also form the basis of planning frameworks and algorithms in literature.
> > >
> > > One case that DAGs cannot represent are parallel agent compositions where agents are interacting with each other to complete the same subgoal. We could treat them as a single agent and analyze their joint behaviors within our current framework. However, it also motivates the harder problem of analysing their behaviors independently and then combining their guarantees.
> > >
> > > Another case where we might not have a clear decomposable structure is when the choice of subtasks and agents are dynamically changing as more outputs are available. This can be modeled by probabilistic edges/transitions between nodes and it motivates a reinforcement learning style problem of picking agents to use at each step. This is also an interesting direction for future work because, as we discuss in our related work section, existing risk-sensitive reinforcement learning frameworks cannot encode the objective that we consider: minimizing the maximum loss incurred which is the correct notion for safety and privacy requirements. Our work is a crucial step towards this more general framework.
> > >
> > > *2. The theoretical analysis seems to rely on several independence and directionality assumptions, as well as implicitly fixed outputs between connected agents. Could the authors clarify which of these are essential for the algorithm’s validity and which could be relaxed?*
> > >
> > > The independence assumption can be relaxed which is evident from the new experiment that we have added where we show that the algorithm is robust to a reasonable amount of correlation between edge losses.
> > >
> > > Directionality and fixed outputs are not a necessary condition. However, we chose this formalization since normally agents are composed by passing the output of one to the next. Nonetheless, our algorithm continues to work when agents, for example, are interacting and modifying a common state/database since we are only analyzing their behaviors.
> > >
> > > In fact, our VaR estimation algorithm can be applied to an arbitrary set of agents that work to complete a task regardless of whether they act sequentially or not. In other words, agents in the same path in the DAG do not need to act sequentially for our algorithm to work.
> > >
> > > *3. The loss function is defined only over the trace T, while the output Y appears not to influence the risk. In realistic settings, the output could affect downstream traces. How would the framework handle such dependencies?*
> > >
> > > This is only a design choice for the formalization of the agent graph however our algorithm is agnostic of whether the output is included within the trace. Indeed, the output can influence subsequent agent behaviors but we believe this correlation is small when the losses quantify objectives like safety, privacy, and fairness.
> > >
> > > We hope that addresses your doubts and concerns! Do let us know if any questions persist.

---

### Official Review · Reviewer_pFjP · 2025-11-01

**Soundness:** 3
**Presentation:** 3
**Contribution:** 3
**Rating:** 4
**Confidence:** 3

**Summary:**

The paper presents a risk-aware framework for optimally selecting agent compositions that are represented as directed acyclic graphs, called agent graphs. The algorithm BucketedVaR utilizes dynamic programming to traverse the agent graph and find the optimal path that minimizes the value-at-risk of losses by dividing the risk budget $\alpha$ among the agents and applying a union bound. Theoretical analysis shows that asymptotically, the quantile estimated by the algorithm is near-optimal when the loss variables along every path are independent or are only loosely correlated, scaling polynomially with the number of agents. The algorithm is tested on a series of reinforcement learning benchmarks to evaluate its effectiveness in identifying optimal paths and quantifying tail behaviors.

**Strengths:**

- This work connects risk-sensitive optimization and compositional agent systems using directed acyclic graphs. The presented BucketedVaR algorithm is polynomial in the number of agents, thereby avoiding exponential enumeration (as seen in baselines).

- The topic presented is relevant and has significance for measuring risk sensitivity in safety-critical AI agent composition. The experiments in various RL benchmarks demonstrate that BucketedVaR successfully identifies the same optimal path as the baseline algorithm across all benchmarks, while providing tight estimates of VaRα.

- The paper is clearly written and well-organized; the agent graph formalism and examples presented help illustrate the ideas effectively.

**Weaknesses:**

- In real AI agent chains, losses are often correlated via shared context or sequential dependence; the paper should analyze or empirically test how violation of independence could affect performance.


- Since the work only tested on RL benchmarks but described LLM examples as potential use cases, missing validation on LLM agentic pipelines where sampling cost and judge noise matter is concerning. For LLMs, losses (e.g., amount of hallucinated information) may be subjective, noisy, or non-numeric. Additionally, black-box sampleability may be significantly more expensive than in RL, where thousands of samples per edge can be easily obtained.


- VaR ignores tail severity and is not a coherent risk measure; the authors should justify this choice of using VaR versus CVaR or provide comparative results.

**Questions:**

1. How sensitive are the algorithm’s guarantees/empirical performance to deviation from independence and to noise in loss measurements?
2. Why was VaR chosen over CVaR?
3. How would the method adapt when only a few samples per agent are feasible, such as LLM-based agents?

---

> ### Author Response · Authors · 2025-11-19
>
> We thank the reviewer for their detailed comments and suggestions. In this first revision, we have made two additions:
>
> 1. **Computation of CVaR**: As most of the reviewers pointed out, conditional value-at-risk, CVaR, is a more useful risk measure as it is sensitive to the tail risk distribution and is a coherent risk measure. We show how we can obtain an approximation of CVaR directly from the computations made by our algorithm and also demonstrate empirically that these approximations are tight on our benchmarks.
>    Since the conditional value-at-risk is the expected tail quantile, it can be approximated by the average of the tail quantiles already computed by our algorithm. Figure 2(b) shows that this approximated CVaR is very close to the baseline CVaR computed by explicit path enumeration.
> 2. **Robustness to correlations in losses**: We implement a new benchmark where we can control the amount of correlation between edge losses and show that our algorithm is robust to a reasonable amount of correlation (up to $\\rho \= 0.5$).
>    This benchmark consists of a synthetic agent graph with a path of varying length where the losses along all the edges are distributed according to correlated Gaussians. We conduct experiments on this benchmark by using our algorithm to approximate the VaR and CVaR while varying the correlation level and path length. As seen in Figure 4, the approximation breaks down under complete correlation but is still robust to some correlation.
>
> Additions in the text are marked by a red font.
>
> Here are detailed responses to the questions raised:
>
> 1. **Correlated/dependent losses**: The additional experiment that we have incorporated demonstrates that our algorithm is robust to some amount of correlation. Beyond this, we believe it is hard to do better without explicit path enumeration in the black-box setting.
> 2. **CVaR as a baseline**: We have included the explicit evaluation of CVaR across all paths as a baseline in our benchmarks. Results are included in Table 2\. We agree with the reviewer that CVaR is a better risk measure that accounts for tail severity. This is why we additionally show how our existing algorithm can also be used to compute the CVaR.
> 3. **Sample complexity**: While we agree that $10^4$ samples can be expensive for LLM/robotic agents, we argue that we cannot estimate the tail quantiles accurately without a reasonably big sample size in the black-box setting. However, as we remark in our conclusion, if our only objective is to find the optimal path without accurately determining its VaR, then online optimization using a multi-armed bandits-style framework would be an interesting future direction.

---

### Official Review · Reviewer_B2Qs · 2025-11-01

**Soundness:** 3
**Presentation:** 3
**Contribution:** 3
**Rating:** 6
**Confidence:** 3

**Summary:**

The paper formalises risk-sensitive composition of agentic workflows represented as DAGs, where path loss is the max over edge-level losses. It proposes BucketedVaR, a dynamic-programming algorithm that allocates a risk budget across edges via a union bound to upper-bound the path value at risk and select a path in polynomial time. Theoretical results include a finite-sample guarantee based on DKW and an asymptotic near-optimality bound under independence of edge losses. Experiments on compositional rl tasks show tight empirical coverage and agreement with an exhaustive baseline on small graphs.

**Strengths:**

1. Clear formulation & motivation: Risk of max loss is appropriate for safety/privacy violations in composed systems.

2. Simple, scalable idea: Union-bound budgeting with DP avoids path enumeration; complexity $O\left(n(d+1)^2|V|^2\right)$.

3. Theory with interpretable slack: Asymptotically, the selected path is within an $\alpha^2 / 2$ quantile-level slack under independence (Thm. 2)

4. Empirical evidence: Tight coverage across benchmarks; agreement with exhaustive baseline; clear sensitivity to buckets/sample.

**Weaknesses:**

1. Choice of VaR over CVaR: VaR ignores tail severity; CVaR is often preferred for safety. The paper mentions CVaR as future work but gives no partial result or empirical check.

2. Missing baselines and ablations. No comparison to chance-constrained shortest path or to dependence-aware relaxations. No experiments that explicitly vary tail dependence (e.g., shared noise seeds, correlated disturbances) to show robustness/failure modes.

3. Finite-sample conservativeness is unclear. Thm. 1 shows the returned threshold q satisfies coverage $\geq 1-(\alpha+\gamma)$ (Eq. 5), i.e., it provides an upper bound on $\text{VaR} _{\alpha+\gamma}$, not necessarily on $\operatorname{VaR} _\alpha$. Hence $q$ can underestimate the true $\operatorname{VaR} _\alpha$ when $\gamma>0$-problematic for safety guarantees. The text claims the estimate is "at least as large as the true VaR" asymptotically, but the finite-sample statement doesn't ensure one-sided conservativeness at level $\alpha$.

4. Graphs are small (e.g., 16 paths), and a is relatively large ( $\geq 0.05$ ). Safety practice often targets $\alpha \leq 10^{-3}$. Sample sizes of $10^4$ per comparison are heavy and may be infeasible for LLM/robotic agents. There is no runtime/scaling analysis on denser DAGs or rarer tails.

**Questions:**

Q1 Dependence-robust, conservative guarantees. How can BucketedVaR be modified to deliver non-anti-conservative VaR ${ }_\alpha$ guarantees under unknown edge-loss dependence without path enumeration?

Q2 What is the empirical sensitivity to tail dependence? Please inject controlled correlation between edge losses (e.g., shared disturbance processes, comonotonic sampling) and report coverage error and path changes versus your baseline.

Q3. Some suggestions for baselines: (i) chance-constrained shortest path surrogates; (ii) CVaR of max; (iii) an independence-aware exact path solver (using order statistics / product CDFs) where feasible.

---

> ### Author Response · Authors · 2025-11-19
>
> We thank the reviewer for their detailed comments and suggestions. In this first revision, we have made two additions:
>
> 1. **Computation of CVaR**: As most of the reviewers pointed out, conditional value-at-risk, CVaR, is a more useful risk measure as it is sensitive to the tail risk distribution and is a coherent risk measure. We show how we can obtain an approximation of CVaR directly from the computations made by our algorithm and also demonstrate empirically that these approximations are tight on our benchmarks.
>    Since the conditional value-at-risk is the expected tail quantile, it can be approximated by the average of the tail quantiles already computed by our algorithm. Figure 2(b) shows that this approximated CVaR is very close to the baseline CVaR computed by explicit path enumeration.
> 2. **Robustness to correlations in losses**: We implement a new benchmark where we can control the amount of correlation between edge losses and show that our algorithm is robust to a reasonable amount of correlation (up to $\\rho \= 0.5$).
>    This benchmark consists of a synthetic agent graph with a path of varying length where the losses along all the edges are distributed according to correlated Gaussians. We conduct experiments on this benchmark by using our algorithm to approximate the VaR and CVaR while varying the correlation level and path length. As seen in Figure 4, the approximation breaks down under complete correlation but is still robust to some correlation.
>
> Additions in the text are marked by a red font.
>
> Here are detailed responses to the questions raised:
>
> 1. **Correlated/dependent losses**: The additional experiment that we have incorporated demonstrates that our algorithm is robust to some amount of correlation. Beyond this, we believe it is hard to do better without explicit path enumeration in the black-box setting.
> 2. **Non-conservative bounds**: The reviewer rightly points out that our algorithm might underestimate the VaR as seen in Theorem 1\. We remark that in practice due to the conservative estimate provided by the union bound, our algorithm tends to produce conservative VaR estimates as seen in all our benchmarks. Alternatively, one can always pick the conservative sample quantile given by the DKW inequality at each edge to obtain a theoretical $1-\\alpha$ coverage guarantee. However, we observed that these estimates were always too loose when combined with the conservativeness of the union bound.
> 3. **CVaR as a baseline**: We have included the explicit evaluation of CVaR across all paths as a baseline in our benchmarks. Results are included in Table 2\. We agree with the reviewer that CVaR is a better risk measure that accounts for tail severity. This is why we additionally show how our existing algorithm can also be used to compute the CVaR.
> 4. **Sample complexity**: While we agree that $10^4$ samples can be expensive for LLM/robotic agents, we argue that we cannot estimate the tail quantiles accurately without a reasonably big sample size in the black-box setting. However, as we discuss in our conclusion, if our only objective is to find the optimal path without accurately determining its VaR, then online optimization using a multi-armed bandits-style framework would be an interesting future direction for study.
> 5. **Chance-constrained stochastic shortest path (CCSSP) \[1\] algorithms as a baseline**: While CCSSP is a related planning framework, we believe that the problem that we consider cannot be encoded or approximated as an instance of CCSSP. As discussed in our related work section, CCSSP along with other existing planning and reinforcement learning frameworks consider optimization of cumulative rewards along paths and cannot reasonably encode the max or min objective that we consider.
> 6. **Independence-aware exact path solver**: We are unsure which methods the reviewer is referring to here and we request them to provide a reference.
>
> \[1\] An Anytime Algorithm for Chance Constrained Stochastic Shortest Path Problems and Its Application to Aircraft Routing. Hong et al. ICRA 2021\.

---

### Author Response · Authors · 2025-12-02
**Final author remarks**

Dear ACs and Chairs,

We first summarize the main strengths of our work as recognized by the reviewers:

- **Clearly motivated and well organized** (B2Qs, pFjP, ZHyK).
- **Simple scalable idea**: Union-bound budgeting with DP avoids path enumeration (B2Qs, pFjP).
- **Tight empirical coverage and agreement with exhaustive baseline** (B2Qs, pFjP).

Next, we discuss how the additional experiments we included in the revised submission address the main questions. Note that additions in the text are marked by a red font.

1. **Computation of CVaR**: Two reviewers (B2Qs, pFjP) suggested that the conditional value-at-risk (CVaR) is a useful risk measure to consider since it is sensitive to tail severity and is a coherent risk measure.
   We show how we can easily approximate CVaR using values precomputed by our algorithm (since CVaR is the expected tail loss, we can approximate it as the average of tail quantiles which are computed by our algorithm).
   Our experiments, shown in Figure 2(b), demonstrate that this approximation of CVaR is tight similar to VaR.
2. **Robustness to correlations in losses**: All the reviewers mention that the independence assumption between edge losses could be restrictive in practice. While this independence assumption is required for our theoretical results, we conduct additional experiments to show that, in practice, our algorithm is robust to reasonable levels of correlation.
   As per the suggestion of Reviewer B2Qs, we create a synthetic benchmark where we can control the level of correlation. This benchmark consists of an agent graph with a path of varying length where the losses along all the edges are distributed according to correlated Gaussians. We conduct experiments on this benchmark by using our algorithm to approximate the VaR and CVaR while varying the correlation level and path length. As seen in Figure 4, the approximation breaks down under complete correlation but is still robust up to a correlation level of 0.7.

Lastly, here are responses to some common questions of the reviewers:

1. **High sample complexity**: It is not possible to estimate the tail quantiles accurately without a reasonable sample size in the restricted black-box setting that we study. However, as discussed in our conclusion, if our only objective is to find the optimal path without accurately determining risk measures, then online optimization using a multi-armed bandits style framework would be an interesting direction for study.

2. **Directionality and output passing between agents**: Directionality and fixed outputs are not a necessary condition for our algorithm to work. However, we chose this formalization since agents are usually composed by passing the output of one to the next. Nonetheless, our algorithm continues to work when agents, for example, are interacting and modifying a common state/database since we are only analyzing their behaviors. In fact, our algorithm can be applied to an arbitrary set of agents that work to complete a task regardless of whether they act sequentially or not. In other words, agents in the same path in the DAG do not need to act sequentially for our algorithm and guarantees to hold.

Sincerely,
Authors of Submission 3827

---

### Meta-Review · Area_Chair_xZPm · 2025-12-29

**Summary:**

The paper proposes a method to quantify the risk associated with agentic systems. The formulations rests on representing agentic systems as a graph of a composition of agents that generate inference trajectories in a given environment. For a given trajectory, the proposed method calculates the value-at-risk (VaR), as well as the conditional value-at-risk (CVaR) for the revised version, using a new algorithm called BucketedVaR. BucketedVaR leverages dynamic programming to approximate the VaR for agent graphs that generate execution trajectories. The authors provide definitions of their risk minimization objective as well as proofs of the bounds of the estimation of BucketedVaR as well as the convergence of BucketedVaR under independent trajectories. The paper also outlines experiments that the empirical performance of BucketedVaR for estimating the risk objective, including a synthetic for somewhat correlated trajectories in the revised version.

The reviewers generally positively noted the importance of the research and its application to agentic systems, as well as the detailed formulations and proofs related to BucketedVaR. The reviewers also positively noted the presentation of the papers, including both the mathematical formulations and empirical results.

The main criticisms of the paper pertain including additional results on CVaR, which are included in the revised version. Another criticism relate to the algorithm and proofs having strict assumptions for when they would be applicable. The authors partially address with a new synthetic experiments that study correlated Gaussian distributions in the agentic loss functions along with discussion of limitations. While the authors addressed important criticism of the reviews, the paper could be further improved by integrating the insights from the discussion along limitations and sample complexity into the final draft.

**Reviewer Concerns:**

Adressed Concerns:
* Reviewer B2Qs' concerns on CVaR and robustness to correlation.
* Reviewer pFjP's concerns on CVaR and robustness to correlation. The concern related to sample complexity is clarified in a comment, and should be included in the final version of the paper.
* Reviewer ZHyK's questions on different results presented in the paper and some of the assumptions of the algorithm.


Outstanding Concerns
* Reviewer B2Qs' clarification on which baselines are applicable.
* Reviewer ZHyK's concerns on the applicability of the algorithm to more complex problems - this can be addressed by further revisions in the limitations section.

**Reviewer Scores:**

* Reviewer B2Qs score remains 6.
* Reviewer pFjP raises to 6.
* Reviewer ZHyK score remains 6.

---

### Decision · Program_Chairs · 2026-01-26

Accept (Poster)